# POSITION: THE HIDDEN COSTS AND MEASUREMENT GAPS OF REINFORCEMENT LEARNING WITH VERIFIABLE REWARDS

**Aaron Tu**[1*]   **Weihao Xuan**[2*]   **Heli Qi**[3*]   **Fang Wu**[6*]   **Xu Huang**[4]   **Qingcheng Zeng**[5]
**Shayan Talaei**[6]   **Yijia Xiao**[7]   **Peng Xia**[8]   **Xiangru Tang**[9]   **Yuchen Zhuang**[4]
**Bing Hu**[10]   **Hanqun Cao**[11]   **Wenqi Shi**[12]   **Tianang Leng**[13]   **Rui Yang**[14]
**Yingjian Chen**[10]   **Ziqi Wang**[15]   **Irene Li**[2]   **Nan Liu**[14]   **Huaxiu Yao**[8]
**Li Erran Li**[16,17]   **Ge Liu**[18]   **Amin Saberi**[6]   **Naoto Yokoya**[2]   **Jure Leskovec**[6]
**Yejin Choi**[6]

[1]University of California, Berkeley   [2]The University of Tokyo   [3]Waseda University
[4]Georgia Institute of Technology   [5]Northwestern University   [6]Stanford University
[7]University of California, Los Angeles   [8]University of North Carolina at Chapel Hill
[9]Yale University [10]Independent Researcher   [11]The Chinese University of Hong Kong
[12]University of Texas Southwestern Medical Center   [13]University of Pennsylvania
[14]National University of Singapore   [15]Liverpool University   [16]AWS AI, Amazon
[17]Columbia University   [18]University of Illinois at Urbana-Champaign

## ABSTRACT

Reinforcement learning with verifiable rewards (RLVR) is a practical, scalable way to improve large language models on math, code, and other structured tasks. However, we argue that many headline RLVR gains are not yet validated because they conflate true policy improvement with three recurring confounds: (i) *budget mismatch* between RLVR and base-model evaluation, (ii) *calibration/attempt inflation* that converts abstentions into confident (sometimes incorrect) answers, and (iii) *data contamination* in legacy benchmarks. Using matched-budget reproductions and partial-prompt contamination audits, we find that several celebrated gaps shrink substantially or disappear under parity-controlled, clean evaluation. These effects do not imply RLVR is ineffective; rather, they show that current reporting often overstates capability gains and obscures reliability costs. We therefore propose a compact, *tax-aware* minimum standard for RLVR training and evaluation that co-optimizes correctness, grounding, and calibrated refusal, and that requires budget parity, variance disclosure, judge-robustness probes, and explicit provenance screening. Our position is constructive: RLVR is effective and deployable for verifiable domains, but "reasoning" gains should be treated as provisional unless a small core of tax-aware controls—budget parity, calibration/abstention tracking, robust evaluation with at least one judge stress test, and a simple contamination audit—is enforced.

## 1 INTRODUCTION

Reinforcement learning with verifiable rewards (RLVR) has become a leading post-training route for improving large language models on math, code, and other structured tasks (Luong et al., 2024; Wen et al., 2025a). By optimizing against automatically computable signals—unit tests for programs, exact numeric or string matches for math, or retrieval-grounded checks for citations—RLVR promises a scalable, label-efficient path to better reasoning. Recent results are striking: across multiple domains, RLVR systems often post large gains on standard benchmarks. Moreover, Figure 2 shows a rise in RLVR-tagged papers alongside improvements on AIME-24/25 through 2024–H1 2025, underscoring both the field's activity context and the need to separate genuine reasoning gains

---

*Equal contribution

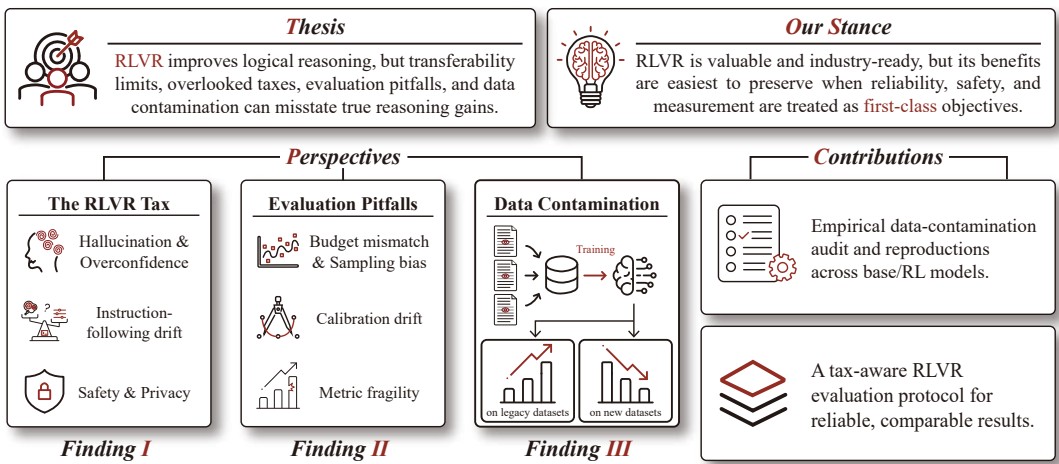

Figure 1: Paper Roadmap: taxes, evaluation pitfalls, contamination, and the unified protocol.

from measurement and budgeting artifacts. Our position is that RLVR is effective and deployable for verifiable domains, but "reasoning" gains should be treated as provisional without a small core of tax-aware controls.

Beneath the surface, however, sits a question that frames the current debate: does RLVR genuinely impart new reasoning capability, or does it mainly sharpen selection among behaviors the base model already knows how to produce? Parity-controlled studies show that base models can narrow or erase RLVR gaps when given matched sampling budgets—consistent with smarter search rather than capability expansion (Yue et al., 2025; Wu et al., 2025a). At the same time, several regimes report gains that are hard to recover by sampling alone: explicitly optimizing the multi-sample objective (*pass@k* training) (Wang et al., 2025), curriculum via self-play with variational problem synthesis (Zhang et al., 2025a), and distribution-aware reward shaping that counters rank bias and diversity collapse (Liu et al., 2025b), alongside longer-horizon RL schedules and unlikeliness rewards (Liu et al., 2025a; He et al., 2025a). Even metric choice can flip conclusions: answer-only scores may diverge from process-aware metrics that require both a correct answer and a valid chain of thought.

A second complication is what we call the *RLVR tax*: unintended *empirical* side effects that ride along with apparent gains under current reasoning-style post-training—reduced abstention, miscalibration, instruction-fidelity drift, and a larger safety/privacy surface due to longer traces. This tax is not mathematically inevitable and not unique to RLVR; similar patterns arise under reasoning-heavy SFT and RLHF. We focus on RLVR because verifiable objectives and open weights make these trade-offs measurable at relatively low cost. In practice, RLVR tends to reduce abstention and increase stated confidence—sometimes even when answers are wrong—thereby shifting risk from "I don't know" to assertive errors (Song et al., 2025; Mei et al., 2025; Yao et al., 2025). It can also chip away at instruction fidelity on longer generations, where adhering to formats or constraints becomes harder as chains grow (Fu et al., 2025; Li et al., 2025). Finally, longer, more explicit reasoning traces expand the attack and leakage surface, raising jailbreak success and privacy exposure if left unchecked (Zhou et al., 2025; Jiang et al., 2025; Ackerman & Panickssery, 2025; Green et al., 2025; Huang et al., 2025; Zhang et al., 2025b).

A third, orthogonal complication is *measurement*. Reported advances are sensitive to sampling budgets and decoding settings, to the stability of LLM-as-a-judge evaluators, to calibration drift, and to data provenance (Hochlehnert et al., 2025; Chandak et al., 2025; Brown et al., 2024; Muennighoff et al., 2025; Zhao et al., 2025b; Wang et al., 2024a;b; Leng et al., 2024; Shen et al., 2025; Wu et al., 2025b; He et al., 2025b; Mirzadeh et al., 2024; Liu et al., 2024b). When budgets are matched, judges are stress-tested, and datasets are versioned and decontaminated, reported improvement gaps can diminish, suggesting that part of the apparent progress reflects evaluation design rather than durable capability (Hochlehnert et al., 2025; Chandak et al., 2025). To help orient the reader, Figure 1 summarizes three threads (taxes, evaluation pitfalls, and contamination) and previews the tax-aware protocol we adopt.

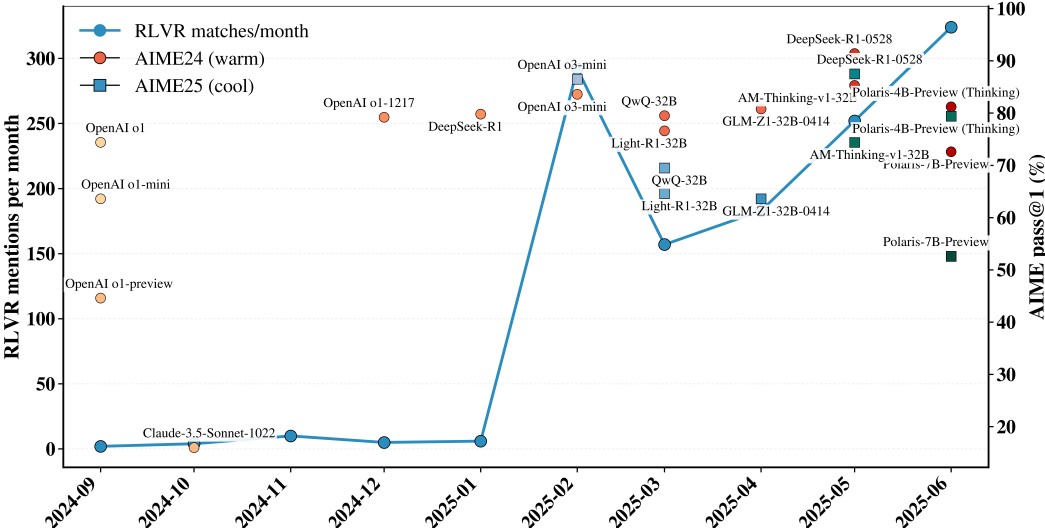

Figure 2: Monthly RLVR activity vs. AIME performance (time span: May 2024–June 2025). **Left axis**: count of pages per month whose *title or abstract* contains "RLVR" or "reinforcement learning with verifiable rewards" (Google Scholar and arXiv via SerpAPI). **Right axis**: *pass@1* (%) on AIME-24 (circles, warm) and AIME-25 (squares, cool) for selected models; labels show model names.

**Scope and definitions.** Throughout, we use *RLVR* to mean post-training that optimizes LLMs against *automatically checkable* signals rather than human preference models (Ouyang et al., 2022). Typical implementations adopt PPO/GRPO/DAPO-style updates with a KL penalty to the base policy, often mixing offline rollouts with online sampling, adding entropy regularization, and using selective filtering or gating. Rewards are frequently *componentized*—for correctness, grounding/citation sufficiency, and calibrated refusal—and introduced in stages. Our analysis centers on open-weight models fine-tuned for math/code/question answering (QA) with verifiable objectives, in both single- and multi-domain regimes. We deliberately avoid a catalogue of low-level algorithmic variants except where they bear on reliability (e.g., unlikeliness rewards that counter rank bias, or curricula that change variance). The primary threats considered are privacy leakage from long chains of thought (CoT) and increased jailbreak susceptibility during or after RLVR.

**Position, Contributions, and Roadmap.** RLVR is valuable and industry-ready, but we encourage prioritizing reliability, safety, and measurement. Our position is that headline reasoning gains should be treated as provisional unless a small core of tax-aware controls is in place. Our contributions are threefold: *(1) Measurement:* we provide parity-controlled reproductions and a unified gap table that isolates inflation sources from budget mismatch, judge and template drift, and dataset versioning (Table 2). *(2) Tax:* we quantify attempt inflation and the resulting calibration and refusal costs under RLVR and matched SFT controls (Table 1). *(3) Protocol:* based on these findings, we distill a compact *tax-aware minimum standard* for RLVR training and evaluation, intended as a minimum credible reporting bar with a default instantiation in section 6, not another broad checklist. The paper is organized accordingly: we (i) synthesize evidence on sharpening vs. expansion (section 2); (ii) analyze the RLVR tax and how it distorts reported improvements (section 3); (iii) reproduce representative gaps under parity controls (section 4); (iv) audit contamination with partial-prompt probes (section 5); and (v) present the tax-aware minimum standard that co-optimizes accuracy, grounding, and calibrated abstention while standardizing budgeting and provenance checks (section 6).

## 2 WHAT CHANGES UNDER RLVR? AN EVIDENCE OVERVIEW

We review empirical signals about RLVR's effects without taking a side on the "expansion *vs.* sharpening" question. Skeptical and optimistic perspectives recur in the literature, and each tends to appear under identifiable conditions.

## 2.1 Skeptical Lens: Sharpening and Sampling Effects

Under a skeptical lens, RLVR often appears to improve sample efficiency and steer the model toward high-reward regions already present in the base distribution, rather than broaden fundamental capability (Gandhi et al., 2025; Shah et al., 2025). When base models are evaluated under matched sampling budgets (i.e., *pass@k*), gaps between base and RLVR-trained models often shrink (Yue et al., 2025), consistent with selection rather than acquisition. Empirically, outputs can collapse toward a dominant pretraining mode with reduced diversity (Zhao et al., 2025a), and theory suggests that after RLVR, support *shrinkage* tends to outweigh support *expansion* (Wu et al., 2025a). Stress tests add caution: performance collapses at higher complexities have been reported even with ample compute (Shojaee et al., 2025), although some failures trace to evaluation artifacts such as token limits and unsolvable items, underscoring the need for budget parity, seed control, and dataset hygiene (see Section 4) (Opus & Lawsen, 2025). Finally, several works separate *knowledge injection* from *policy optimization*: pipelines that use SFT or distillation to inject knowledge and RLVR to select among candidates typically beat pure GRPO, suggesting RLVR optimizes within learned support (Ma et al., 2025; Wan et al., 2025; Chen et al., 2025a); guidance and self-distillation further help over vanilla GRPO (Nath et al., 2025; Liu et al., 2025c).

## 2.2 Optimistic Lens: Signals of Capability Broadening

Under an optimistic lens, carefully designed RLVR appears to *expand* what models can do, not merely sharpen selection. Prolonged optimization with explicit KL control and periodic resets (ProRL) reports trajectories that are not reachable by sampling the base policy alone, with the largest gains where the base initially struggles (Liu et al., 2025a). Reward shaping also matters: by diagnosing a GRPO tendency to reinforce high-probability completions, He et al. (2025a) show that adding unlikeliness rewards and training deeper widens the model's support and improves multi-sample theorem proving, indicating qualitative changes to reachable reasoning modes. These effects are conditioned by pretraining coverage: cross-domain analyses find broader benefits in Math/Code/Science, while under-represented areas such as Logic/Simulation/Tabular realize meaningful gains when RLVR is applied in-domain (Cheng et al., 2025). Consistent with this, process-aware metrics that require both a correct answer and a syntactically valid chain (CoT-*pass@k*) often reveal larger RLVR deltas than answer-only measures (Wen et al., 2025b). While generalization to substantially harder regimes remains an open goal (Sun et al., 2025b), the weight of evidence supports a constructive view: with sufficient optimization depth, bias-aware rewards, and the right domain coverage, RLVR can broaden reasoning capabilities.

**Reporting note.** Answer-only metrics (e.g., *pass@k*) can show muted gains, whereas process-aware criteria such as CoT-*pass@k*—which require both a correct answer and a syntactically valid chain—often reveal larger deltas in settings explicitly optimized for process rewards (Wen et al., 2025b). We treat these as *behavioral* signals rather than explanations: chains need not be faithful to internal computation (Chen et al., 2025c). Accordingly, we report both answer- and process-aware metrics alongside calibration (ECE/abstention), and we prefer budget-matched saturation curves over single-point *pass@k* (Section 4).

**Cross-Lens Lessons:** Across both perspectives, two themes are consistent: *(i)* the base model's strength and coverage strongly condition observed gains, and *(ii)* reward/metric design and evaluation protocol (sampling budgets, seeds, and contamination hygiene) can change the sign of conclusions. These observations motivate our stance: treat reliability, safety, and measurement as first-class objectives. The following sections detail the RLVR tax (Section 3), show how evaluation choices inflate or erode claims (Section 4), and present a unified, tax-aware training and evaluation protocol that yields more trustworthy estimates and more transferable improvements (Section 6).

## 3 Perspective I — The RLVR Tax: Benefits, Costs, and Controls

RLVR reliably lifts accuracy on verifiable tasks, but those gains are easy to misread if we ignore systematic costs—the *RLVR tax*. By "RLVR tax" we mean empirical side effects observed under current reasoning-style post-training regimes: reduced abstention, miscalibration and overconfidence, drift in instruction fidelity on long generations, and a larger safety/privacy surface due to longer and more

Table 1: Factual-QA control tracking Abstention (↓), Shared Accuracy (↑), and ECE (↓). *Interpretation.* "Not attempted" is the *absolute* number of items with no extractable answer, computed on the block-specific evaluation set for that family (totals differ across families). At 14B/32B, reasoning SFT sharply reduces abstentions but leaves shared accuracy flat and worsens calibration; at a larger scale (DeepSeek), RL reduces abstention and improves shared accuracy and ECE, largely by *attempting more items*. Accuracy on *newly attempted* tail items is modest.

| Family/Scale | Model (version) | Not attempted ↓ | Accuracy (shared, %) ↑ | ECE (shared) ↓ |
|---|---|---|---|---|
| Qwen2.5 14B | Qwen2.5-14B-Instruct | 1136 | 12.5 | 0.598 |
| | R1-Distill-Qwen-14B (SFT) | 102 | 10.5 | 0.692 |
| | RL-Reasoning-14B | 103 | 10.0 | 0.684 |
| Qwen2.5 32B | Qwen2.5-32B-Instruct | 2492 | 17.4 | 0.591 |
| | R1-Distill-Qwen-32B (SFT) | 76 | 17.5 | 0.640 |
| | RL-Reasoning-32B | 63 | 17.1 | 0.600 |
| DeepSeek | DeepSeek-V3 (Instruct) | 480 | 27.5 | 0.496 |
| | DeepSeek-R1 (RL) | 81 | 34.6 | 0.317 |

explicit traces. This tax is not mathematically inevitable and not unique to RLVR; similar patterns can arise under reasoning-heavy SFT and RLHF. We center RLVR because verifiable objectives and open weights make these trade-offs particularly measurable.

We focus on three recurrent pressures: *(i)* hallucination and overconfidence, *(ii)* erosion of instruction following, and *(iii)* safety and privacy exposure. Each maps directly to our broader thesis: without tax-aware training and sober measurement, reported "reasoning gains" are easy to overstate and hard to transfer.

## 3.1 HALLUCINATION AND OVERCONFIDENCE

RLVR can suppress refusals while amplifying confident errors. Empirically, refusal rates often collapse after RLVR, shifting abstentions into assertive answers (Song et al., 2025); models may also repeat flawed steps or produce CoT that diverges from the final answer even when accuracy rises (Yao et al., 2025). A plausible mechanistic reading is that sparse, verifiable rewards combined with entropy pressure tend to encourage determinacy under weak evidence, producing high-variance gradients and spurious local optima (Li & Ng, 2025). Consistent with this, multiple studies find substantial miscalibration: self-reported confidence remains high even on *incorrect* responses and can increase with longer CoT traces (Mei et al., 2025; Zeng et al., 2025; Kirichenko et al., 2025).

We make this concrete with a factual-QA control that tracks *Not attempted* (lower is better), *Accuracy (shared)*—accuracy on items that both models at a given scale attempted (higher is better)—and *Expected Calibration Error (ECE; lower is better)* computed on shared items from the model's stated confidence. The dominant effect is *attempt inflation*: stronger RLVR policies attempt far more questions. For example, moving from **DeepSeek-V3** to **DeepSeek-R1** (Guo et al., 2025) sharply reduces abstentions and improves shared ECE, but accuracy on the *newly attempted* tail is modest, so reported gains depend on how one aggregates across attempted vs. unattempted items (Zeng et al., 2025). In other words, scaling RL *encourages attempts*; whether this yields net quality improvements depends on calibration and gating.

Formally, let $A_{\text{base}}$ and $A_{\text{RL}}$ denote the sets of items for which the base and RLVR models produce an extractable answer, and let $A_{\cap} = A_{\text{base}} \cap A_{\text{RL}}$ be their intersection. We report "shared accuracy" and ECE on $A_{\cap}$, and define "attempt inflation" as the number of additionally attempted items

$$\left| A_{\text{RL}} \setminus A_{\text{base}} \right|,$$

i.e., questions the RLVR model answers that the base model leaves unattempted. The dominant effect in Table 1 is attempt inflation: $|A_{\text{RL}} \setminus A_{\text{base}}|$ grows sharply, while accuracy on $A_{\cap}$ changes only modestly.

## 3.2 Instruction Following

Optimizing hard for reasoning can erode controllability—especially for long generations. Several studies observe regressions in instruction fidelity when training emphasizes extended chain-of-thought or purely verifiable endpoints (Fu et al., 2025; Li et al., 2025). Moreover, recent evidence shows that models tend to overfit to a small set of verifiable constraints and struggle to generalize; IFBench (58 out-of-domain constraints) documents this gap and finds that RLVR improves precise instruction-following generalization (Pyatkin et al., 2025). This is why our minimum standard in Section 6 assumes either an explicit instruction or format component in the reward, or at least a separate instruction-following evaluation pack.

## 3.3 Safety and Privacy

Long, explicit reasoning traces widen the attack and leakage surface. Frontier reasoning models are jailbreakable at high rates; automated attacks approach deterministic success in the studied settings, and success rises with attempt and context budgets (Kassianik & Karbasi, 2025; Zhou et al., 2025; Jiang et al., 2025). Many-shot evaluations show the same scaling. Targeted defenses tuned specifically to that regime can sharply reduce measured attack success in controlled tests, but these gains are brittle across models and budgets and can impose utility costs (Ackerman & Panickssery, 2025). Richer CoT also increases privacy risk: attributes, confidential prompts, and dataset contents can be reconstructed more easily as intermediate reasoning grows (Green et al., 2025). Safety tuning itself can impose a "safety tax," degrading math and coding unless staged and balanced carefully (Huang et al., 2025; Zhang et al., 2025b).

> **Threat Model (Reasoning Traces).** *Who sees traces:* exposure risk is highest for open-weight models or deployments that return full CoT; API-only hidden traces lower user-facing risk, while internal logs can aid auditing. *Training vs. inference:* RLVR increases trace length and attempt budgets during training and (unless capped) inference, which can amplify leakage/jailbreak surfaces. *What "risk increase" means:* higher probability that sensitive prompt/data fragments or unsafe behaviors appear in controllable intermediate text. *When traces help:* hidden/internal traces improve monitorability and post-hoc safety audits, so the goal is controlled exposure, not blanket removal.

**Takeaway:** These taxes distort reported gains in predictable ways: overconfident hallucinations inflate apparent utility, instruction drift undermines deployability, and expanded attack surfaces raise real-world risk. The answer is not to abandon RLVR but to *co-optimize* for reliability—compose rewards so correctness, grounding, and abstention can all improve; manage variance and difficulty; and make calibration and provenance part of evaluation. We follow this recipe in the next sections to show where conclusions change once the tax accounted for in (section 4, section 6).

## 4 Perspective II — Pitfalls in Evaluation: Are We Measuring Real Reasoning Gains Accurately?

Claims about RLVR progress are highly sensitive to sampling budgets, metric design, and dataset hygiene. This matters for our thesis: budget mismatch, fragile judge pipelines, and calibration drift can overstate "reasoning gains" and complicate transfer beyond math and code. We treat contamination separately in Section 5 and focus here on budgeting, metric robustness, and calibration.

## 4.1 Budget Parity and Saturation

A recurring pattern in the literature is to report *pass@k* for RLVR models while holding base models to much smaller budgets (e.g., *pass@1* or *pass@5*). In such settings, measured improvements often reflect extra search rather than a better policy. Prior work stresses matched budgets and decontaminated baselines for fair comparison (Wen et al., 2025a; Wu et al., 2025b). Small or underspecified benchmarks inflate estimator variance and reduce statistical power, making single-run point estimates unstable; averaging across multiple randomized trials materially improves estimate reliability (Mu et al., 2025). Under parity controls (e.g., *SoberScore*, a standardized, matched-budget, multi-seed evaluation that fixes decoding/setup and reports mean ± std), several prominent gaps

shrink or disappear (Hochlehnert et al., 2025). In practice, evaluations should match $k$ across base and RLVR, plot saturation curves (accuracy vs. $k$, optionally summarized by area under the curve), and disclose decoding budgets and parameters.

## 4.2 METRIC FRAGILITY AND LLM JUDGES

While RLVR often targets verifiable domains (math/coding), many evaluation targets either lack programmatic verifiers or have only partial/weak ones—e.g., safety/refusal appropriateness, long-form coherence and rationale quality, multi-turn task success, or fuzzy information extraction. In those cases, we fall back to LLM-as-a-judge. LLM-judge pipelines offer convenience but can be brittle: small changes—seed choice, $k$, dataset versioning, instruction placement, option order, or even tensor-parallel settings—produce swings comparable to reported gains (Sun et al., 2025a). Judges are also manipulable (Zhao et al., 2025b). Beyond judges, seemingly modest metric or setup shifts can alter conclusions about "stable reasoning" (Liu et al., 2024a). Where programmatic verifiers exist, they should be preferred. When judges are unavoidable, prompt/format perturbations and adversarial probes should be part of the protocol, with robustness deltas reported, inter-judge agreement documented, and all templates/configurations released to reduce hidden degrees of freedom.

## 4.3 CALIBRATION DRIFT AND OVERCONFIDENCE

RL optimization often sharpens the output distribution. Top-1 accuracy can rise even as calibration worsens, increasing brittleness under distribution shift (Hochlehnert et al., 2025). Combined with the volatility noted above (Sun et al., 2025a), some apparent "wins" likely reflect confident exploitation of evaluation quirks rather than robust reasoning. Evaluations should therefore track expected calibration error (ECE), output entropy, and refusal/abstention alongside accuracy, and consider early stopping or annealing when calibration degrades even if reward continues to increase.

Table 2: Reported scores vs. standardized evaluation (*avg@32* of *pass@1*, estimated by averaging 32 independent single-sample decodes) with matched decoding budgets and a shared verifier/prompt family. $\Delta$ denotes Reported $-$ Standardized. See Appendix 4 for comprehensive standardized results of recent RLVR models.

| Model (checkpoint) | Benchmark | Reported (setup) | Standardized Eval | $\Delta$ |
|---|---|---|---|---|
| Nemotron-Research-Reasoning-Qwen-1.5B v1 | AIME-24 | 48.13 (*pass@1*; Liu et al. 2025a) | 45.62 | +2.51 |
| | AIME-25 | 33.33 (*pass@1*; Liu et al. 2025a) | 33.85 | −0.52 |
| | AMC-23 | 79.29 (*pass@1*; Liu et al. 2025a) | 85.70 | −6.41 |
| | Math | 91.89 (*pass@1*; Liu et al. 2025a) | 92.01 | −0.12 |
| | Minerva | 47.98 (*pass@1*; Liu et al. 2025a) | 39.27 | +8.71 |
| | Olympiad | 60.22 (*pass@1*; Liu et al. 2025a) | 64.56 | −4.34 |
| AceReason-Nemotron-14B | AIME-24 | 78.60 (*avg@64*; Yang et al. 2025) | 77.29 | +1.31 |
| | AIME-25 | 67.40 (*avg@64*; Yang et al. 2025) | 66.04 | +1.36 |
| DAPO-Qwen-32B | AIME-24 | 50.00 (accuracy; Yu et al. 2025) | 51.56 | −1.56 |
| Open-RS3-1.5B | AIME-24 | 46.70 (*pass@1*; Dang & Ngo 2025) | 30.94 | +15.76 |
| STILL-3-1.5B | AIME-24 | 39.33 (*avg@64*; $T$=0.6, top-$p$=0.95; Min et al. 2025) | 31.46 | +7.87 |
| DeepScaleR-1.5B | AIME-24 | 43.10 (*pass@1*; Luo et al. 2025) | 38.54 | +4.56 |
| Polaris-7B-Preview | AIME-24 | 72.60 (*avg@32*; NLP & collaborators 2025) | 66.46 | +6.14 |
| | AMC-23 | 89.00 (*avg@8*; NLP & collaborators 2025) | 93.59 | −4.59 |

## 4.4 REPORTED VS. REPRODUCED: A GAP ANALYSIS

To make these issues concrete, we compared widely cited checkpoints to parity-controlled runs using the same verifier, matched decoding budgets, and aligned dataset versions. The gaps in Table 2 arise from three main factors:

Sampling budgets. Multi-sample reporting (e.g., *avg@64* with $T$=0.6, top-$p$=0.95 for STILL-3-1.5B) inflates scores relative to single-shot *pass@1* under the same verifier (Min et al., 2025). In our standardized runs this manifests as +7.87 for STILL and +6.14 for Polaris-7B on AIME-24 (NLP & collaborators, 2025).

Template and version drift. Changes in prompt templates and dataset slices (e.g., AIME curation, AMC answer formats) shift accuracy by several points, with exact hashes frequently missing from

Table 3: Partial–prompt contamination summary. The Math block (left) reports answer–match accuracy at an 80% prefix on two legacy sets vs. the fresh AIME–2025. The SimpleQA block (right) is a non–math control, where QWEN's advantage largely attenuates. Full results, including ACC@60/40 and ROUGE–L/EM for all models and datasets, are in Tables 6–10 in the Appendix.

| Model | Math (ACC@80) | | | SimpleQA | |
|---|---|---|---|---|---|
| | MATH-500 | AMC-23 | AIME-2025 | R@80 | EM@80 |
| Llama-3-1.8B | 2.8 | 0.0 | 0.0 | 37.11 | 19.86 |
| Qwen2.5-Math-7B | 58.0 | 52.5 | 0.0 | 29.34 | 12.47 |
| Qwen2.5-32B | 60.0 | 52.5 | 3.3 | 41.06 | 23.09 |
| Qwen3-14B-Base | 58.2 | 47.5 | 0.0 | 37.24 | 19.40 |

model cards (NLP & collaborators, 2025; Liu & contributors, 2025). This helps explain mixed signs for Nemotron-1.5B v1 (Minerva $+8.71$ vs. AMC-23 $-6.41$).

Metric and decision rules. Sources sometimes mix "accuracy," *pass@k*, *avg@k* (mean of *pass@1* across draws), and *maj@k* (majority vote), or substitute judges where verifiers exist. Once we normalize to a single verifier/spec, several deltas compress (e.g., DAPO-32B on AIME-24: $-1.56$) (Sun et al., 2025a; Zhao et al., 2025b).

Finally, small-set variance matters: on AIME-24/25, seed and decode settings alone can yield $\pm3$–5 percentage points. Confidence intervals and saturation curves are needed to separate noise from effect (Mu et al., 2025; Hochlehnert et al., 2025).

**Why this matters**  Under unmatched budgets, unprobed judges, and untracked calibration, distributional *sharpening* can masquerade as capability *expansion*. Clean, parity-controlled evaluation often shrinks or flips celebrated gains. Hence our checklist—budget parity and saturation curves, variance disclosure, judge robustness, calibration metrics, and a contamination audit (Section 5)—is essential to distinguish real progress from artifacts and to make the *tax-aware* protocol credible.

## 5    PERSPECTIVE III — DATA CONTAMINATION

Data provenance is a first-order confound: if pretraining or RL data overlap evaluation sets, measured "reasoning" may reflect memorization. Contamination-aware evaluations in coding (Liu et al., 2024b) and multi-domain leaderboards (Liang et al., 2023) already recommend strict provenance checks; our partial-prompt audit provides direct evidence in math. Partial-prompt reconstruction is a high-precision but not exhaustive signal of overlap: we treat it as one useful probe to be triangulated with lexical and fuzzy matching and with fresh test sets.

**Probe design (partial-prompt completion).**  Following Wu et al. (2025b), we reveal only the first $x\%$ of each problem ($x \in \{80, 60, 40\}$), greedily decode the suffix, and compute ROUGE-L@$x$, EM@$x$, and ACC@$x$. High EM/ACC under large prefixes indicates the model can reconstruct the hidden tail and final answer verbatim.

**Case study: QWEN vs. LLAMA.**  As summarized in Table 3, QWEN2.5–MATH–7B and QWEN3–14B–BASE achieve high ACC@80 on legacy math sets (around 58–60% on MATH-500) yet collapse on AIME–2025 (0–3%). LLAMA–3–1.8B remains near zero across the same sets, consistent with clean evaluation. This pattern—strong on older math sets, absent on the newest release—is consistent with substantial overlap between legacy math benchmarks and the training corpora of some QWEN-family models, and with comparatively cleaner evaluation for LLAMA. (See App. Tables 6–9 for ACC/ROUGE/EM at 80/60/40% prefixes.) Similar "contamination-free" stance and methodology are advocated in code by Liu et al. (2024b) and in broader evaluations by Liang et al. (2023).

**SimpleQA control.**  On a non–math control (SIMPLEQA) there is no systematic QWEN advantage: for example, QWEN2.5–32B attains 41.06/23.09 (R@80/EM@80) versus LLAMA–3–1.8B at 37.11/19.86 (App. Table 10). The attenuation on an unseen domain supports the interpretation that elevated partial–prompt math scores on legacy sets reflect contamination rather than a general

suffix–reconstruction ability. We use SimpleQA as a non-math control to check that strong tail-reconstruction behavior does not appear uniformly across domains; we do not assume SimpleQA itself is free of contamination.

**Controls and implications.** We (i) treat contaminated sets as *probes* of memorization, not reasoning; (ii) prioritize uncontaminated or freshly released test sets; and (iii) publish prompts/seeds/configurations to enable third-party screening. When we re-evaluate RLVR vs. base models under these controls, widely cited gaps shrink or flip. This directly supports our thesis: contamination can make modest distributional sharpening appear as frontier expansion.

## 6 THE TAX-AWARE MINIMUM STANDARD FOR RLVR

Our empirical results in Sections 4 and 5 show that three pieces of measurement infrastructure materially change how RLVR gains appear: parity-controlled budgeting, calibration and attempt tracking, and contamination audits. Alongside synthesized prior work on judge robustness and safety evaluations, we package these into a *tax-aware minimum standard* for reporting RLVR results. The standard is intentionally narrow and measurement-focused. It does not prescribe a particular RL algorithm. Instead, it specifies the controls that must be in place before headline reasoning gains are treated as reliable.

**(1) Budget parity and saturation curves.** First, RLVR models and their baselines must be evaluated under matched sampling budgets. In our gap analysis, several celebrated improvements shrank or disappeared once we fixed the verifier, prompt family, decoding parameters, and number of samples per item (Table 2). Reporting *pass@k* for the RLVR model and *pass@1* for the base model, or silently changing temperature, top-$p$, or stopping rules, makes it impossible to tell how much of the gain comes from extra search rather than a better policy.

Under the minimum standard, any claim that RLVR improves a metric on a given benchmark must include: (i) the exact sampling budget for both base and RLVR models, (ii) a saturation curve that plots accuracy as a function of $k$ under a shared decoding setup, and (iii) at least three seeds with mean and confidence intervals or standard deviations (Mu et al., 2025; Hochlehnert et al., 2025). We recommend summarizing performance by area under the saturation curve in addition to a single *pass@k* point, since this is less sensitive to one particular choice of $k$ and reveals whether RLVR shifts the whole budget–performance frontier or only improves at very large $k$.

**(2) Calibration, abstention, and judge robustness.** Second, evaluations must track calibration and abstention, not only accuracy. Section 3 showed that RLVR often reduces refusals and increases stated confidence, which yields more attempted items but also more confident errors. Our factual QA control (Table 1) separates *shared* accuracy from *newly attempted* tail items and measures expected calibration error (ECE) on shared items. This reveals a distinct failure mode: headline scores rise because the model stops saying "I do not know" and starts answering everything, while accuracy on overlapping items barely moves.

Under the minimum standard, each reported accuracy figure must be accompanied by: (i) refusal or "not attempted" rates, (ii) shared accuracy on the intersection of items both models attempted, and (iii) a calibration metric such as ECE computed from the model's confidence scores (Hochlehnert et al., 2025; Leng et al., 2024). For settings that rely on LLM-as-a-judge rather than programmatic verifiers, we also require at least one *judge stress test*: the same outputs scored under several prompt templates or instruction orderings, with the spread in scores reported (Zhao et al., 2025b; Sun et al., 2025a). This does not remove judge fragility, but it makes visible how sensitive a claimed RLVR gain is to small changes in the judge pipeline. Together, these measurements ensure that gains are not driven primarily by attempt inflation, miscalibration, or a brittle judge configuration.

**(3) Contamination audits and data hygiene.** Third, RLVR claims must be supported by explicit data provenance checks. Section 5 used partial-prompt completion to show that some math benchmarks are heavily memorized by QWEN families: the model can reconstruct the hidden tail and final answer when given 80 percent of the problem on legacy sets, but not on fresh ones such as AIME-2025. Without such checks, apparent reasoning gains can simply reflect better recall of training data (Liu et al., 2024b; Liang et al., 2023; Wu et al., 2025b).

Under the minimum standard, any benchmark used to support an RLVR claim must be accompanied by: (i) a contamination screen that combines fuzzy or lexical matching against pretraining and fine-tuning corpora (where available) and partial-prompt probes at several prefix lengths, and (ii) at least one clean held-out set that shows no evidence of tail reconstruction even at large prefixes. For benchmarks that do show contamination, we treat them as *probes* of memorization rather than as primary evidence of reasoning and clearly label them as such. We also require that dataset versions, prompt templates, and filtering rules be released or described precisely enough for others to re-run the audit.

**Summary.** The tax-aware minimum standard asks for three things before treating RLVR gains as robust: matched budgets with saturation curves and variance disclosure, calibration and abstention metrics with at least one judge stress test when judges are used, and a contamination audit with at least one clean held-out benchmark. Our experiments show that each component can change conclusions about whether RLVR expands capability or sharpens selection. Taken together, these controls provide a simple, concrete bar that future RLVR work can meet without specifying a particular training recipe, and that can be reused for SFT, RLHF, and test-time compute evaluations.

# 7 CONCLUSION

Our central position is that RLVR is effective and deployable for verifiable domains, but headline "reasoning" gains should be treated as provisional unless a small core of tax-aware controls—budget parity, calibration/abstention tracking, robust evaluation with at least one judge stress test, and a simple contamination audit—is enforced. RLVR delivers real gains on verifiable tasks, but the field often over-indexes on headline accuracy while under-weighting *taxes* (hallucination/overconfidence, instruction drift, safety & privacy exposure) and *measurement* (budget mismatch, judge fragility, calibration drift, contamination). Under parity controls and calibration tracking, several celebrated "reasoning gains" shrink, suggesting that part of the progress reflects distributional *sharpening* rather than durable expansion.

**Broader relevance.** The same pressures arise in SFT, RLHF, and test-time compute (e.g., CoT and *maj@k*). This tax-aware minimum standard applies unchanged beyond RLVR.

**Why focus on RLVR.** We center RLVR because programmatic verifiers make the measurement problem tractable and reveal where gains come from. As a result, it is a useful *proving ground* for reliability methods that transfer to broader LLM training and deployment. We hope this synthesis helps the community separate genuine capability expansion from artifacts of budgets, metrics, and data, and encourages tax-aware training and reporting across RLVR, SFT, and RLHF.

## ETHICS STATEMENT

This paper raises some potential ethics concerns under the ICLR Code of Ethics and describes the steps we took to mitigate them:

**Privacy & leakage.** Long, explicit chains-of-thought can reveal sensitive content or training artifacts. We evaluate only on public math/code benchmarks and do not use private prompts. For any illustrative traces, we sanitize identifiers and redact potentially sensitive strings. Our standardized evaluation caps best-of-$N$ and CoT length to reduce leakage surfaces.

**Dual-use: jailbreaks & misuse.** Methodological details about jailbreak stress tests could be re-purposed to increase harm. We report aggregate metrics and robustness deltas, but do not release attack payloads or automation scripts that would materially increase misuse without corresponding defenses. Our evaluation protocol co-optimizes calibrated abstention alongside accuracy.

**Measurement integrity.** Selective budgeting, fragile judge pipelines, or metric switching can mis-represent results. To reduce this risk, we use matched decoding budgets, fixed verifier/prompt families, multi-seed averaging, and release configurations (decoding parameters, seeds, templates) sufficient to reproduce our reported numbers.

## REPRODUCIBILITY STATEMENT

We make our results reproducible by fixing verifiers and a shared prompt family (section 4), matching decoding budgets and reporting decoding parameters, and averaging over three seeds (Table 4, Table2; Figure 2). Datasets and methodology for contamination screens and judge-robustness probes referenced in section 5 are provided with instructions to re-run the audits. Hardware details and aggregate compute ($128 \times$H100 (96 GB) cluster and $\approx$3,500 GPU-hours) are included to support cost and throughput replication.

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

# A  APPENDIX

**Conventions and metrics.**  $pass@k$ = probability at least one of $k$ samples is correct; $avg@k$ = mean $pass@1$ across $k$ draws; $maj@k$ = majority vote over $k$; $ECE$ = expected calibration error. Arrows ($\uparrow/\downarrow$) indicate whether higher or lower is better.

**Usage of Large Language Models.**  We used an LLM assistant as a productivity tool for light editing and formatting: smoothing wording the authors had already written, polishing captions, refactoring LaTeX tables/macros, and drafting small utility snippets (e.g., plotting and CSV loaders) that we then modified and verified. All research ideas, evaluation designs, hyperparameters, and analysis decisions are by the authors, and every experiment and number reported was run and checked by us.

# B  POTENTIAL CRITICISMS AND RESPONSES

**On capability expansion.**  RLVR can expand capability under specific regimes (e.g., prolonged RL with resets and KL control; unlikeliness-based shaping; domain-targeted RLVR with sparse pretraining) (Liu et al., 2025a; He et al., 2025a; Cheng et al., 2025). Our claim is not negation but *measurement*: without tax-aware training and parity-controlled evaluation, expansion is often overstated. The protocol in §6 is compatible with these positive regimes while curbing overclaiming.

**On "it's just data."**  Data quality and curricula matter (Chen et al., 2025a; Wen et al., 2025a), but reward/metric design independently shapes failure modes (determinacy/overconfidence under judge rewards; terse/off-topic outputs under naive factuality; calibration drift) (Chen et al., 2025b; Leng et al., 2024; Hochlehnert et al., 2025). We treat *both* data and objectives as first-class knobs (see §6).

**On budgeted *pass@k*.**  We use *pass@k* for practical relevance, but ask for *matched budgets* and *saturation curves*. Under these controls, headline gaps often shrink or flip (Hochlehnert et al., 2025). Reporting AUC alongside *pass@k* keeps comparisons fair (§4).

**On safety/privacy risk.**  Jailbreak success scales with attempts/context; many-shot tuning can reduce measured rates in controlled settings (Zhou et al., 2025; Jiang et al., 2025; Ackerman & Panickssery, 2025). Longer CoT increases leakage surfaces (Green et al., 2025). Our protocol caps Best-of-$N$/CoT length at eval and co-optimizes abstention/privacy with accuracy (§6).

**On reliance on LLM judges.**  When verifiers are unavailable or partial, we use LLM judges with robustness probes (prompt/format perturbations, adversarial tests) and publish templates/configs; judges are manipulable, so robustness deltas are reported (Zhao et al., 2025b; Sun et al., 2025a). Prefer verifiers when possible (§4).

**Scope.**  Our synthesis centers on verifiable math/code/QA with open-weight models. Agentic/multimodal settings introduce additional privacy/safety channels and may need tailored verifiers and audits; we expect core principles (budget parity, calibration, abstention, contamination screens) to transfer with domain-specific adaptations.

# C  STANDARDIZED EVALUATIONS

We report full results under matched budgets, fixed verifiers, and a shared prompt family. These tables extend Section 4 and underlie the saturation-curve argument: once $k$, templates, and dataset versions are controlled, several headline gaps narrow. All results in this section use the same verifier and prompt template family; we specify $k$, number of seeds, and dataset versions below.

**Key observations.**  Thinking-mode (test-time compute) delivers large gains across sizes: for 4B models, Qwen3-4B→Qwen3-4B (Thinking) yields +50.5/+44.7 on AIME-24/25 and ≈+28 on the averaged score; Polaris-4B-Preview→Thinking shows +52.6/+54.7 and ≈+29 on Avg; for 8B, Qwen3-8B→Thinking adds +51.9/+46.3 and ≈+28 on Avg. Small RLVR-tuned models can exceed

larger non-thinking baselines: the 1.5B Nemotron-Research-Reasoning v2 averages **61.70**, outscoring non-thinking 4–8B baselines (Qwen3-4B **46.98**, Qwen3-8B **48.96**) and leading them by $\sim$24–30 points on AIME-24. Within size families, leaders are consistent across benchmarks (7B AceReason-1.1 Avg **74.44**; 14B AceReason Avg **77.33**; 32B DeepSeek-R1-Distill Avg **72.53**), while the frontier DeepSeek-R1-0528 tops overall (AIME-24 **90.00**, AIME-25 **78.33**, Avg **81.72**). Several columns (AMC/MATH) cluster in the mid-90s, indicating saturation under our $k$=32/verifier setup. Together, these patterns support (i) test-time compute as an effective lever for reasoning; (ii) RLVR's disproportionate lift at smaller scales; and (iii) the need for matched budgets, response-length controls, and saturation curves to interpret small nominal deltas fairly.

Table 4: Standardized evaluation across math benchmarks (higher is better). *Styling:* within each size family, the best score for a given benchmark is **bold** and the second-best is underlined. **Context:** Parity-controlled scores used in Section 4. **Setup:** *pass@1* estimated by averaging over $k$=32 independent single-sample decodes (*avg@32*), same verifier and prompt family, 3 seeds. **Compute:** evaluations ran on a $128\times$H100 (96 GB) cluster and consumed $\approx$3,500 GPU-hours.

| Model | AIME-24↑ | AIME-25↑ | AMC-23↑ | MATH↑ | Minerva↑ | Olympiad↑ | Avg↑ |
|---|---|---|---|---|---|---|---|
| Qwen2.5-Math-1.5B | 8.33 | 6.35 | 44.06 | 66.67 | 18.42 | 30.74 | 29.10 |
| Qwen2.5-Math-1.5B-Instruct | 10.10 | 8.85 | 55.08 | 74.83 | 29.32 | 40.00 | 36.37 |
| DeepSeek-R1-Distill-Qwen-1.5B | 31.15 | 24.06 | 72.81 | 85.01 | 32.18 | 51.55 | 49.46 |
| STILL-3-1.5B | 31.46 | 25.00 | 75.08 | 86.24 | 32.77 | 53.84 | 50.73 |
| DeepScaleR-1.5B | 38.54 | 30.52 | 80.86 | 88.79 | 36.19 | 58.95 | 55.64 |
| Qwen2.5-Math-1.5B-Oat-Zero | 20.00 | 10.00 | 52.50 | 74.20 | 26.84 | 37.78 | 36.89 |
| Open-RS1 | 30.94 | 22.60 | 73.05 | 84.90 | 29.92 | 52.82 | 49.04 |
| Open-RS2 | 28.96 | 24.37 | 73.52 | 85.06 | 29.74 | 52.63 | 49.05 |
| Open-RS3 | 30.94 | 24.79 | 72.50 | 84.47 | 29.11 | 52.25 | 49.01 |
| Nemotron-Research-Reasoning-Qwen-1.5B v1 | 45.62 | **33.85** | 85.70 | 92.01 | 39.27 | 64.56 | 60.17 |
| Nemotron-Research-Reasoning-Qwen-1.5B v2 | **51.77** | 32.92 | **88.83** | **92.24** | **39.75** | **64.69** | **61.70** |
| Qwen2.5-Math-7B | 15.62 | 6.56 | 52.81 | 67.72 | 15.64 | 32.44 | 31.80 |
| Qwen2.5-Math-7B-Instruct | 12.19 | 9.17 | 58.36 | 83.21 | 35.56 | 41.60 | 40.01 |
| DeepSeek-R1-Distill-Qwen-7B | 53.23 | 38.96 | 89.30 | 93.95 | 43.07 | 66.67 | 64.20 |
| Qwen2.5-Math-7B-Oat-Zero | 26.67 | 6.67 | 67.50 | 79.20 | 32.72 | 41.78 | 42.42 |
| Skywork-OR1-7B | 66.88 | 51.15 | 92.73 | 96.04 | 44.03 | 73.61 | 70.74 |
| LEAD-7B | 51.67 | 37.19 | 89.06 | 93.73 | 43.11 | 66.34 | 63.52 |
| AceReason-Nemotron-7B | 65.83 | 47.19 | **95.08** | 95.81 | **45.35** | 73.92 | 70.53 |
| AceReason-Nemotron-1.1-7B | **71.56** | **64.58** | 93.36 | **96.73** | 44.05 | **76.37** | **74.44** |
| Polaris-7B-Preview | 66.46 | 51.56 | 93.59 | 95.68 | 44.47 | 73.65 | 70.90 |
| Qwen2.5-14B | 11.04 | 7.92 | 47.19 | 73.19 | 22.51 | 37.01 | 33.14 |
| Qwen2.5-14B-Instruct | 13.65 | 12.40 | 58.13 | 80.28 | 38.63 | 43.23 | 41.05 |
| DeepSeek-R1-Distill-Qwen-14B | 67.81 | 48.33 | 95.39 | 95.28 | 46.43 | 72.06 | 70.88 |
| LEAD-14B | 64.06 | 52.29 | 92.81 | 95.23 | 47.52 | 72.25 | 70.69 |
| AceReason-Nemotron-14B | **77.29** | **66.04** | **98.67** | **96.90** | **47.73** | **77.34** | **77.33** |
| Qwen2.5-32B | 15.62 | 9.17 | 59.30 | 76.51 | 26.42 | 41.45 | 38.08 |
| Qwen2.5-32B-Instruct | 17.19 | 14.17 | 67.66 | 83.17 | 40.96 | 47.85 | 45.17 |
| DeepSeek-R1-Distill-Qwen-32B | **69.06** | **55.52** | **95.62** | **95.74** | **46.50** | **72.76** | **72.53** |
| DAPO-Qwen-32B | 51.56 | 36.98 | 92.73 | 80.74 | 33.07 | 48.97 | 57.34 |
| Enigmata-Qwen2.5-32B | 61.67 | 46.88 | 91.25 | 93.69 | 46.32 | 69.14 | 68.16 |
| DeepSeek-V3-0324 | 55.83 | 43.33 | 92.50 | 95.12 | **48.58** | 66.91 | 67.05 |
| DeepSeek-R1-0528 | **90.00** | **78.33** | **99.38** | **97.80** | 48.35 | **76.44** | **81.72** |
| Qwen3-4B | 21.88 | 17.92 | 66.95 | 84.27 | 38.50 | 52.38 | 46.98 |
| Qwen3-4B (Thinking) | 72.40 | 62.60 | 95.78 | 96.31 | 46.44 | 76.40 | 74.99 |
| Polaris-4B-Preview | 27.29 | 23.02 | 71.56 | 85.89 | 39.20 | 59.13 | 51.02 |
| Polaris-4B-Preview (Thinking) | **79.90** | **77.71** | **99.45** | **97.38** | **47.21** | **80.38** | **80.34** |
| Qwen3-8B | 25.42 | 20.21 | 69.61 | 84.54 | 39.81 | 54.15 | 48.96 |
| Qwen3-8B (Thinking) | **77.29** | 66.46 | 95.00 | **96.86** | **49.06** | **77.56** | **77.04** |
| DeepSeek-R1-0528-Qwen3-8B | 75.73 | **67.29** | **97.19** | 96.32 | 45.85 | 74.19 | 76.10 |

# D    SEQUENCE LENGTH AND COMPUTE FOOTPRINT

We summarize generated token lengths by benchmark as a proxy for compute footprint and exposure surface. Longer chains increase latency and may worsen calibration and safety/privacy risks (Section 3), so we report typical generation lengths under the same decoding budgets used in Appx. C.

Table 5: Median generated output length (tokens per problem; answer + reasoning) under the standardized decoding setup. **Context:** Compute/privacy footprint complementing accuracy. **Setup:** Same prompts/verifier as Appx. C; decoding budget $k=32$; lengths aggregated across all samples per problem (correct and incorrect). **Takeaway:** Reasoning-tuned models often produce much longer traces (e.g., 2–8× vs. instruct baselines), which impacts evaluation cost and safety/privacy exposure.

| Model | AIME-24 | AIME-25 | AMC-23 | Math | Minerva | Olympiad | Avg |
|---|---|---|---|---|---|---|---|
| Qwen2.5-Math-1.5B | 1114 | 1033 | 828 | 648 | 959 | 904 | 914 |
| Qwen2.5-Math-1.5B-Instruct | 990 | 891 | 801 | 566 | 640 | 810 | 783 |
| DeepSeek-R1-Distill-Qwen-1.5B | 16363 | 16252 | 9979 | 5700 | 8194 | 11873 | 11394 |
| STILL-3-1.5B | 13350 | 13000 | 7716 | 4314 | 5921 | 9345 | 8941 |
| DeepScaleR-1.5B | 9780 | 8978 | 5003 | 3139 | 5270 | 5807 | 6330 |
| Qwen2.5-Math-1.5B-Oat-Zero | 1166 | 1155 | 848 | 626 | 689 | 892 | 896 |
| Open-RS1 | 13578 | 13698 | 7495 | 4170 | 5935 | 8995 | 8978 |
| Open-RS2 | 14215 | 13618 | 7612 | 4174 | 5805 | 9059 | 9081 |
| Open-RS3 | 14394 | 13606 | 7884 | 4233 | 5688 | 9047 | 9142 |
| Nemotron-Research-Reasoning-Qwen-1.5B | 7786 | 7713 | 6294 | 5070 | 6569 | 6678 | 6685 |
| Qwen2.5-Math-7B | 1201 | 1152 | 946 | 720 | 1136 | 950 | 1018 |
| Qwen2.5-Math-7B-Instruct | 1465 | 1418 | 983 | 670 | 748 | 1051 | 1056 |
| DeepSeek-R1-Distill-Qwen-7B | 13613 | 14543 | 6402 | 4125 | 5595 | 8988 | 8878 |
| Qwen2.5-Math-7B-Oat-Zero | 1032 | 1178 | 884 | 673 | 696 | 870 | 889 |
| Skywork-OR1-7B | 15366 | 17845 | 8361 | 5541 | 8566 | 11818 | 11250 |
| LEAD-7B | 10838 | 11573 | 4863 | 3111 | 3692 | 7054 | 6855 |
| AceReason-Nemotron-1.1-7B | 14331 | 16502 | 6672 | 3835 | 6676 | 9060 | 9513 |
| Polaris-7B-Preview | 12564 | 14389 | 6538 | 4313 | 6125 | 8681 | 8768 |
| Qwen2.5-14B | 1076 | 1088 | 847 | 591 | 1209 | 815 | 938 |
| Qwen2.5-14B-Instruct | 1079 | 994 | 850 | 607 | 632 | 844 | 834 |
| DeepSeek-R1-Distill-Qwen-14B | 11295 | 13389 | 5735 | 3781 | 4919 | 8042 | 7860 |
| LEAD-14B | 8364 | 9019 | 4469 | 3031 | 4675 | 5713 | 5879 |
| AceReason-Nemotron-14B | 13871 | 16334 | 7239 | 4609 | 7677 | 10030 | 9960 |
| Qwen2.5-32B | 1165 | 1055 | 782 | 556 | 1186 | 800 | 924 |
| DeepSeek-R1-Distill-Qwen-32B | 10979 | 13012 | 5826 | 3652 | 4663 | 7924 | 7676 |
| DAPO-Qwen-32B | 6627 | 6074 | 3086 | 2812 | 3887 | 5122 | 4601 |
| Enigmata-Qwen2.5-32B | 13204 | 15417 | 10195 | 6358 | 13770 | 12098 | 11840 |
| DeepSeek-V3-0324 | 3601 | 3474 | 2134 | 1281 | 705 | 2386 | 2264 |
| DeepSeek-R1-0528 | 16276 | 18416 | 10285 | 6041 | 7085 | 13750 | 11976 |
| MiMo-7B-Base | 21716 | 18842 | 15734 | 6390 | 13809 | 13299 | 14965 |
| MiMo-7B-SFT | 13012 | 14250 | 7030 | 4177 | 6152 | 8421 | 8841 |
| MiMo-7B-RL-Zero | 18105 | 18575 | 9950 | 6564 | 6546 | 12152 | 11982 |
| Qwen3-4B | 8909 | 6079 | 2369 | 1289 | 812 | 3217 | 3779 |
| Qwen3-4B (Thinking) | 14750 | 17813 | 8290 | 5076 | 6682 | 10606 | 10536 |
| Polaris-4B-Preview | 5582 | 6148 | 2637 | 1293 | 804 | 4632 | 3516 |
| Polaris-4B-Preview (Thinking) | 28369 | 33109 | 15125 | 9333 | 13307 | 21832 | 20179 |
| Qwen3-8B | 8375 | 6151 | 2863 | 1479 | 662 | 3624 | 3859 |
| Qwen3-8B (Thinking) | 14764 | 18296 | 8756 | 5391 | 7105 | 11384 | 10949 |
| DeepSeek-R1-0528-Qwen3-8B | 20970 | 22742 | 12895 | 7640 | 9329 | 16108 | 14947 |

# E   DATA CONTAMINATION PROBES: MATH

We use partial-prompt completion to detect tail reconstruction: reveal the first $x\%$ of the problem ($x \in \{80, 60, 40\}$), greedily decode the remainder, and score ACC/EM/ROUGE-L at that prefix. High scores at large prefixes indicate memorization of legacy sets rather than reasoning. Tables here expand Section 5.

Table 6: Accuracy of QWEN3 checkpoints on five mathematics benchmarks when each model receives only the first $x\%$ of the question ($x = 80, 60, 40$) and must greedily complete the remainder. Columns "ACC (X%)" report the average accuracy at that prefix length. **Takeaway:** QWEN3 variants achieve high ACC@80 on legacy MATH-500/AMC-23 but collapse on AIME-2025, consistent with contamination in older sets.

| Model | Dataset | ACC (80%) | ACC (60%) | ACC (40%) |
|---|---|---|---|---|
| Qwen3-0.6B-Base | MATH-500 | 26.40 | 14.40 | 5.40 |
| | AMC-23 | 20.00 | 5.00 | 0.00 |
| | AIME 2024 | 3.33 | 0.00 | 0.00 |
| | AIME 2025 | 0.00 | 0.00 | 0.00 |
| | Minerva-Math | 1.10 | 0.00 | 0.00 |
| Qwen3-1.7B-Base | MATH-500 | 33.80 | 18.40 | 9.00 |
| | AMC-23 | 20.00 | 15.00 | 0.00 |
| | AIME 2024 | 0.00 | 0.00 | 0.00 |
| | AIME 2025 | 3.33 | 3.33 | 0.00 |
| | Minerva-Math | 2.94 | 2.21 | 0.74 |
| Qwen3-4B-Base | MATH-500 | 43.20 | 28.20 | 15.00 |
| | AMC-23 | 37.50 | 37.50 | 17.50 |
| | AIME 2024 | 10.00 | 6.67 | 10.00 |
| | AIME 2025 | 3.33 | 0.00 | 0.00 |
| | Minerva-Math | 4.78 | 2.31 | 0.74 |
| Qwen3-8B-Base | MATH-500 | 52.00 | 35.60 | 24.80 |
| | AMC-23 | 35.00 | 25.00 | 27.50 |
| | AIME 2024 | 13.33 | 6.67 | 10.00 |
| | AIME 2025 | 3.33 | 3.33 | 0.00 |
| | Minerva-Math | 5.88 | 3.31 | 1.84 |
| Qwen3-14B-Base | MATH-500 | 58.20 | 44.80 | 30.20 |
| | AMC-23 | 47.50 | 37.50 | 32.50 |
| | AIME 2024 | 10.00 | 16.67 | 16.67 |
| | AIME 2025 | 0.00 | 0.00 | 0.00 |
| | Minerva-Math | 5.88 | 2.21 | 2.94 |
| Qwen3-30B-A3B | MATH-500 | 62.40 | 51.40 | 37.80 |
| | AMC-23 | 40.00 | 25.00 | 25.00 |
| | AIME 2024 | 13.33 | 23.33 | 13.33 |
| | AIME 2025 | 3.33 | 0.00 | 0.00 |
| | Minerva-Math | 4.78 | 3.21 | 1.84 |

Table 7: Accuracy of QWEN2.5 and LLAMA-3 checkpoints on five mathematics benchmarks when each model receives only the first $x\%$ of the question ($x = 80, 60, 40$) and must greedily complete the remainder. **Takeaway:** QWEN2.5 shows strong tail reconstruction on legacy math; LLAMA-3-1.8B remains near zero across sets, reinforcing the contamination interpretation.

| Model | Dataset | ACC (80%) | ACC (60%) | ACC (40%) |
|---|---|---|---|---|
| Qwen2.5-Math-7B | MATH-500 | 58.00 | 44.40 | 29.20 |
| | AMC-23 | 52.50 | 42.50 | 32.50 |
| | AIME 2024 | 16.67 | 20.00 | 16.67 |
| | AIME 2025 | 0.00 | 0.00 | 0.00 |
| | Minerva-Math | 7.72 | 4.41 | 2.94 |
| Qwen2.5-7B | MATH-500 | 44.80 | 27.80 | 15.00 |
| | AMC-23 | 27.50 | 27.50 | 22.50 |
| | AIME 2024 | 6.67 | 0.00 | 3.33 |
| | AIME 2025 | 3.33 | 3.33 | 0.00 |
| | Minerva-Math | 6.62 | 4.78 | 1.47 |
| Qwen2.5-7B-Instruct | MATH-500 | 44.80 | 28.60 | 14.00 |
| | AMC-23 | 40.00 | 15.00 | 12.50 |
| | AIME 2024 | 3.33 | 0.00 | 0.00 |
| | AIME 2025 | 3.33 | 0.00 | 0.00 |
| | Minerva-Math | 10.29 | 5.51 | 3.31 |
| Qwen2.5-14B | MATH-500 | 50.60 | 35.80 | 21.00 |
| | AMC-23 | 40.00 | 27.50 | 27.50 |
| | AIME 2024 | 10.00 | 3.33 | 6.67 |
| | AIME 2025 | 3.33 | 0.00 | 0.00 |
| | Minerva-Math | 8.46 | 6.25 | 2.21 |
| Qwen2.5-32B | MATH-500 | 60.00 | 47.80 | 32.00 |
| | AMC-23 | 52.50 | 45.00 | 42.50 |
| | AIME 2024 | 16.67 | 13.33 | 10.00 |
| | AIME 2025 | 3.33 | 0.00 | 6.67 |
| | Minerva-Math | 9.93 | 4.41 | 2.94 |
| Llama-3-1.8B | MATH-500 | 2.80 | 2.40 | 2.00 |
| | AMC-23 | 0.00 | 0.00 | 0.00 |
| | AIME 2024 | 0.00 | 0.00 | 0.00 |
| | AIME 2025 | 0.00 | 0.00 | 0.00 |
| | Minerva-Math | 1.84 | 0.37 | 0.00 |

Table 8: RougeL and exact-match (EM) scores for QWEN3.0 checkpoints on five mathematics benchmarks when each model receives only the first $x\%$ of the question ($x = 80, 60, 40$) and must greedily complete the remainder. Columns "R@$x$" and "EM@$x$" report the average ROUGE-L and EM for that prefix length. **Takeaway:** Elevated ROUGE-L/EM at large prefixes on legacy sets (but not on AIME-2025) indicates suffix reconstruction rather than emergent reasoning.

| Model | Dataset | R@80 | EM@80 | R@60 | EM@60 | R@40 | EM@40 |
|---|---|---|---|---|---|---|---|
| Qwen3-0.6B-Base | MATH-500 | 53.31 | 20.80 | 45.90 | 6.40 | 39.33 | 1.00 |
| | AMC-23 | 52.26 | 7.50 | 40.39 | 0.00 | 33.92 | 0.00 |
| | AIME 2024 | 54.85 | 20.00 | 27.79 | 0.00 | 24.76 | 0.00 |
| | AIME 2025 | 54.69 | 10.00 | 33.47 | 0.00 | 29.16 | 0.00 |
| | Minerva-Math | 31.41 | 1.47 | 29.17 | 0.00 | 24.86 | 0.00 |
| Qwen3-1.7B-Base | MATH-500 | 55.86 | 24.60 | 49.05 | 10.20 | 41.91 | 2.40 |
| | AMC-23 | 64.35 | 30.00 | 52.57 | 20.00 | 44.95 | 12.50 |
| | AIME 2024 | 56.55 | 26.67 | 36.48 | 6.67 | 36.56 | 6.67 |
| | AIME 2025 | 53.90 | 16.67 | 39.40 | 6.00 | 31.09 | 0.00 |
| | Minerva-Math | 33.99 | 3.31 | 30.32 | 0.00 | 27.28 | 0.00 |
| Qwen3-4B-Base | MATH-500 | 65.04 | 37.40 | 55.98 | 18.80 | 46.44 | 7.00 |
| | AMC-23 | 73.24 | 52.50 | 68.56 | 45.00 | 63.89 | 35.00 |
| | AIME 2024 | 68.98 | 43.33 | 53.22 | 30.00 | 53.22 | 26.67 |
| | AIME 2025 | 55.34 | 10.00 | 39.29 | 0.00 | 27.72 | 0.00 |
| | Minerva-Math | 34.13 | 2.94 | 32.19 | 0.37 | 28.18 | 0.00 |
| Qwen3-8B-Base | MATH-500 | 70.98 | 47.20 | 63.69 | 29.60 | 53.84 | 15.20 |
| | AMC-23 | 78.23 | 57.50 | 69.09 | 52.50 | 69.73 | 47.50 |
| | AIME 2024 | 77.28 | 60.00 | 55.76 | 33.33 | 57.85 | 30.00 |
| | AIME 2025 | 51.97 | 10.00 | 38.60 | 0.00 | 32.21 | 0.00 |
| | Minerva-Math | 35.87 | 2.57 | 33.41 | 0.00 | 28.52 | 0.00 |
| Qwen3-14B-Base | MATH-500 | 74.93 | 55.80 | 70.08 | 39.80 | 60.23 | 23.80 |
| | AMC-23 | 75.32 | 55.00 | 73.20 | 60.00 | 75.52 | 52.50 |
| | AIME 2024 | 72.31 | 50.00 | 61.48 | 40.00 | 60.54 | 40.00 |
| | AIME 2025 | 56.07 | 10.00 | 36.41 | 0.00 | 32.46 | 0.00 |
| | Minerva-Math | 38.12 | 4.41 | 34.11 | 0.74 | 29.72 | 0.00 |
| Qwen3-30B-A3B | MATH-500 | 80.12 | 62.40 | 75.03 | 45.60 | 64.93 | 33.00 |
| | AMC-23 | 80.40 | 60.00 | 73.24 | 55.00 | 77.21 | 55.00 |
| | AIME 2024 | 74.26 | 53.33 | 66.53 | 40.00 | 62.90 | 33.33 |
| | AIME 2025 | 51.57 | 6.67 | 43.18 | 0.00 | 30.55 | 0.00 |
| | Minerva-Math | 37.01 | 3.68 | 34.96 | 1.10 | 29.22 | 0.00 |

Table 9: RougeL and exact-match (EM) scores for QWEN2.5 and LLAMA-3 on five mathematics benchmarks when each model receives only the first $x\%$ of the question ($x = 80, 60, 40$) and must greedily complete the remainder. **Takeaway:** QWEN2.5 shows strong reconstruction on legacy math; LLAMA-3 remains low.

| Model | Dataset | R@80 | EM@80 | R@60 | EM@60 | R@40 | EM@40 |
|---|---|---|---|---|---|---|---|
| Qwen2.5-Math-7B | MATH-500 | 79.63 | 61.60 | 70.30 | 41.20 | 60.81 | 25.40 |
| | AMC-23 | 77.35 | 57.50 | 71.47 | 50.00 | 66.29 | 42.50 |
| | AIME 2024 | 69.35 | 53.33 | 55.90 | 30.00 | 56.53 | 30.00 |
| | AIME 2025 | 48.20 | 10.00 | 36.31 | 0.00 | 27.23 | 0.00 |
| | Minerva-Math | 33.42 | 2.94 | 30.41 | 0.37 | 26.88 | 0.00 |
| Qwen2.5-7B | MATH-500 | 66.09 | 38.20 | 58.15 | 18.00 | 48.34 | 6.00 |
| | AMC-23 | 69.47 | 47.50 | 67.70 | 45.00 | 61.08 | 35.00 |
| | AIME 2024 | 61.28 | 30.00 | 46.19 | 16.67 | 46.14 | 16.67 |
| | AIME 2025 | 54.62 | 10.00 | 42.41 | 0.00 | 30.56 | 0.00 |
| | Minerva-Math | 33.88 | 2.94 | 31.13 | 0.00 | 27.06 | 0.00 |
| Qwen2.5-7B-Instr | MATH-500 | 60.05 | 28.60 | 51.10 | 10.40 | 43.41 | 2.60 |
| | AMC-23 | 61.38 | 30.00 | 51.22 | 17.50 | 42.90 | 10.00 |
| | AIME 2024 | 55.41 | 20.00 | 36.28 | 0.00 | 30.64 | 0.00 |
| | AIME 2025 | 58.12 | 10.00 | 37.96 | 0.00 | 29.81 | 0.00 |
| | Minerva-Math | 34.37 | 2.94 | 28.50 | 0.00 | 25.99 | 0.00 |
| Qwen2.5-14B | MATH-500 | 69.50 | 45.00 | 61.20 | 27.00 | 51.24 | 12.80 |
| | AMC-23 | 74.58 | 47.50 | 68.20 | 42.50 | 67.82 | 37.50 |
| | AIME 2024 | 65.42 | 40.00 | 49.76 | 23.33 | 48.97 | 23.33 |
| | AIME 2025 | 52.56 | 6.67 | 38.02 | 0.00 | 32.92 | 0.00 |
| | Minerva-Math | 35.64 | 3.68 | 31.30 | 0.00 | 27.93 | 0.00 |
| Qwen2.5-32B | MATH-500 | 76.98 | 56.60 | 71.14 | 41.20 | 58.32 | 25.20 |
| | AMC-23 | 79.18 | 62.50 | 74.59 | 60.00 | 75.78 | 55.00 |
| | AIME 2024 | 73.25 | 56.67 | 60.92 | 40.00 | 60.47 | 33.33 |
| | AIME 2025 | 53.67 | 6.67 | 40.91 | 0.00 | 31.90 | 0.00 |
| | Minerva-Math | 34.88 | 2.94 | 32.92 | 0.00 | 28.32 | 0.00 |
| Llama-3-1.8B | MATH-500 | 46.49 | 14.00 | 38.75 | 3.00 | 32.19 | 0.60 |
| | AMC-23 | 40.46 | 0.00 | 29.87 | 0.00 | 25.28 | 0.00 |
| | AIME 2024 | 49.70 | 16.67 | 31.45 | 0.00 | 27.13 | 0.00 |
| | AIME 2025 | 50.16 | 6.67 | 32.97 | 0.00 | 27.18 | 0.00 |
| | Minerva-Math | 35.32 | 1.47 | 29.17 | 0.00 | 26.95 | 0.00 |

# F DATA CONTAMINATION PROBES: SIMPLEQA (CONTROL)

To test whether partial-prompt reconstruction reflects a general suffix-completion ability, we repeat the probe on SIMPLEQA. If math-set effects were general, we would expect similarly high ROUGE-L/EM at large prefixes here. They are not: scores cluster modestly across families.

Table 10: Rouge-L and exact-match (EM) scores on SIMPLEQA when each model receives only the first $x\%$ of the question ($x = 80, 60, 40$) and must greedily complete the remainder. **Takeaway:** No consistent QWEN advantage; effects seen on legacy math do not generalize, supporting the contamination interpretation.

| Model | Dataset | R@80 | EM@80 | R@60 | EM@60 | R@40 | EM@40 |
|---|---|---|---|---|---|---|---|
| Qwen2.5-Math-7B | SimpleQA | 29.34 | 12.47 | 16.19 | 0.69 | 13.28 | 0.00 |
| Qwen2.5-7B | SimpleQA | 37.61 | 19.63 | 19.88 | 1.85 | 14.64 | 0.23 |
| Qwen2.5-7B-Instruct | SimpleQA | 36.06 | 17.78 | 19.76 | 1.62 | 15.87 | 0.46 |
| Qwen2.5-14B | SimpleQA | 39.97 | 21.94 | 21.69 | 3.23 | 15.98 | 0.23 |
| Qwen2.5-32B | SimpleQA | 41.06 | 23.09 | 21.87 | 3.23 | 14.78 | 0.00 |
| Llama-3-1.8B | SimpleQA | 37.11 | 19.86 | 20.24 | 2.08 | 14.22 | 0.00 |
| Qwen3-0.6B-Base | SimpleQA | 26.45 | 10.39 | 16.35 | 0.46 | 14.42 | 0.00 |
| Qwen3-1.7B-Base | SimpleQA | 27.55 | 12.47 | 16.50 | 0.92 | 14.97 | 0.23 |
| Qwen3-4B-Base | SimpleQA | 33.52 | 16.40 | 17.87 | 1.85 | 14.93 | 0.00 |
| Qwen3-8B-Base | SimpleQA | 33.99 | 16.40 | 19.23 | 1.39 | 15.39 | 0.00 |
| Qwen3-14B-Base | SimpleQA | 37.24 | 19.40 | 21.44 | 2.31 | 16.65 | 0.23 |
| Qwen3-30B-A3B-Base | SimpleQA | 38.80 | 19.63 | 22.07 | 3.00 | 16.55 | 0.00 |
| Qwen3-23B-A22B-Instr-2507 | SimpleQA | 53.88 | 38.34 | 34.58 | 10.39 | 22.74 | 1.39 |

