# OpenReview forum: "Position: The Hidden Costs and Measurement Gaps of Reinforcement Learning with Verifiable Rewards"
_ICLR.cc/2026/Conference — ICLR 2026 Conference Withdrawn Submission_

### Official Review · Reviewer_Atx3 · 2025-10-23

**Soundness:** 3
**Presentation:** 3
**Contribution:** 2
**Rating:** 4
**Confidence:** 3

**Summary:**

This paper provides a perspective on unseen costs associated with large performance gains reported when reinforcement learning with variable rewards (RLVR) is used to enhance large language models focused on math, coding, and other structured tasks. The authors argue that some of these gains might be overstated, and lay out a three-pronged thesis to support their claim. They propose a tax-aware RLVR protocol that is claimed to mitigate the identified pitfalls of RLVR.

**Strengths:**

(+) The paper is generally very well-written, and the authors lay out their case for the proposed protocol very nicely.

(+) The paper presents `both sides' of the RLVR coin and provides perspective on the impacts of RLVR by examining recent literature. This perspective is given in terms of the potential of RLVR expanding model capabilities, improving generalizability, and whether RLVR indeed improves sample efficiency.

(+) The premise of an RLVR tax is very interesting- in that the authors posit that gains and advances reported when RLVR is used must take into consideration associated systemic costs- akin to a tax.

(+) The authors highlight the pitfalls of mismatched budgets and the brittleness of metrics when reporting performance of models/ methods in the literature. This is an especially important observation in a field where hundreds of papers appear daily, each claiming superior performance over current art.

(+) The authors make a convincing case for development of a tax-aware RLVR protocol by highlighting challenges in evaluation standards, contaminated data, and unseen/ unaccounted costs associated with large performance gains.

(+) The compositionality and ease of interpretability of the protocol reward makes it amenable for widespread use.

**Weaknesses:**

(-) Analysis is limited in its scope to open-weight models, and further, fine-tuned for math/ code/ QA with verifiable objectives.

(-) The notion of a `tax' in this paper is somewhat limited in its scope to discussions on RLVR. In terms of the three ``recurrent pressures'' mentioned by the authors in Sec. 3 (Lines 190-193), this reviewer is curious as to whether such an interpretation (i.e., costs associated with huge performance gains of a technology) has been more widely applied to the examination of performance and capabilities of LLMs as a broader field, or even to other sub-domains beyond RLVR.

(-) The takeaway from Sec. 3 seems very generic (Lines 265-269) in that such a co-optimization strategy to improve reliability while maintaining provenance is a solution approach that can be broadly used across multiple domains beyond RLVR. Similarly, in Sec. 4 and 5, the importance of standardizing evaluation procedures and impacts of contaminated data in terms of distributional shifts has been widely known across multiple experimental domains (not limited to LLMs and computer science).

(-) Perhaps the most significant weakness in my opinion is that the main body of the paper contains very little results to support the claims made in Sec. 3, 4, 5. Many of the associated results have been deferred to the appendix. Moreover, the proposed protocol itself has not been explicitly evaluated (I might be mistaken in this observation, so authors please correct me if I'm wrong); rather the authors point to work in the literature where it is claimed to work. In this case, the paper amounts to providing a reward model as in Line 444-445 for the working of mechanisms described in paragraph (F) of Sec. 6.

**Questions:**

Please see Weaknesses above.

---

> ### Author Response · Authors · 2025-11-28
> **Response to reviewer Atx3**
>
> Thank you for the careful and balanced review. We appreciate both your summary of what you found useful and your concerns about scope and evidence.
>
> On the scope (open weights; math, code, and QA with verifiable objectives), this restriction is intentional but not sufficiently foregrounded in the current draft. In the revision we will state up front that our claims are strongest exactly in regimes where we can:
>
> - run standardized verifiers;
> - control budgets and decoding pipelines; and
> - run contamination screens,
>
> which currently means open weight, verifiable tasks. We will also add a short discussion of how the same reliability questions extend to closed weight and agentic settings, while being honest that we cannot audit those systems to the same depth.
>
> On the "tax" framing and generality of the co optimization story, we agree that many of the ideas (co optimizing reliability with performance, standardizing evaluation, worrying about contamination) apply well beyond RLVR. Our updated position will say this explicitly: we view RLVR as a natural proving ground for these issues, not as the only place they arise. We will soften any language that suggests the tax framing is novel on its own, and instead emphasize what is RLVR specific here: the combination of verifiable objectives, test time compute, and open checkpoints that makes the measurement problem tractable.
>
> On the lack of visible main text evidence and the protocol not being directly evaluated, we agree that the current structure hides too much in the appendix and over states what has been "shown". In the revision we will:
>
> - move the key quantitative pieces (parity controlled gap analysis, attempt and calibration behavior, partial prompt contamination contrasts) into the main body; and
> - present the protocol as a minimum reporting standard distilled from those observations plus prior work, not as an ablated, optimized training recipe.
>
> Where we only have indirect support (for example for some of the more detailed scheduling suggestions), we will frame those as hypotheses and pointers for future empirical work rather than as claims.
>
> We hope this makes clearer how we see the paper: not as introducing a new RLVR algorithm, but as using RLVR to surface and structure a set of reliability and measurement issues that we expect to matter broadly in post training.

---

### Official Review · Reviewer_cCAc · 2025-11-01

**Soundness:** 4
**Presentation:** 3
**Contribution:** 3
**Rating:** 6
**Confidence:** 3

**Summary:**

This is a good position paper on Reinforcement Learning with Verifiable Rewards (RLVR). It reveals three factors often overlooked in RLVR-related work: the RLVR tax, evaluation traps, and data contamination. It also reevaluates the performance of RLVR-related work under the same experimental settings and points out potentially viable solutions for each factor.

**Strengths:**

- This work points out aspects that researchers in the RLVR field may have overlooked, preventing reported results from being achieved at the expense of other performance metrics
- The authors evaluated the results of mainstream models using the same, fair, and reasonable experimental settings, which helps establish a more equitable evaluation understanding for researchers in the field

**Weaknesses:**

- In Section 6, this tax-aware RLVR protocol proposes several methods that can mitigate or resolve various RLVR tax issues. However, these are largely based on findings from existing work and lack experimental validation under the unified perspective presented in this paper. The practical effectiveness of these solutions is therefore questionable.
- The experiments in this paper mainly focus on data contamination, performance, and model output behavior, lacking a direct evaluation of impacts on privacy exposure and fine-grained instruction-following capability

**Questions:**

See weaknesses part

---

> ### Author Response · Authors · 2025-11-28
> **Response to reviewer cCac**
>
> Thank you for the assessment and for emphasizing both what works and what feels undersupported in the current draft.
>
> On the concern that Section 6 is largely literature driven and insufficiently validated as a whole, we agree. Our intention is not to claim that the entire protocol has been ablated end to end, but to:
>
> - condense what our own measurements say about budget parity, calibration and attempt behavior, and contamination in RLVR; and
> - propose a minimal reporting standard that is consistent with that evidence and with recent practice.
>
> In revision we will be explicit about this boundary. We will mark which components are directly grounded in our empirical sections, and which are forward looking suggestions synthesized from existing work rather than claims of a proven recipe.
>
> On the lack of direct experiments on privacy exposure and fine grained instruction following, we agree this is a real gap. At the moment we mostly reason from others' results plus our own measurements of attempt inflation and calibration. We do not have the space or compute to add a full privacy or instruction following benchmark suite, so instead we will:
>
> - add a concise threat model box that makes our safety and privacy claims concrete and scoped; and
> - soften the language in Sections 3 and 6 to avoid implying that we have already demonstrated reductions in privacy risk or instruction following regressions under our protocol.
>
> We will also point more clearly to existing work that directly measures those axes and position our protocol as a way to incorporate such metrics, rather than as something whose benefits on those dimensions are already empirically established.
>
> Overall, we see your comments as a push to be clearer about where we are diagnosing current practice versus where we are proposing a unified lens, and we will adjust the paper to reflect that.

---

### Official Review · Reviewer_rvSS · 2025-11-01

**Soundness:** 3
**Presentation:** 2
**Contribution:** 3
**Rating:** 6
**Confidence:** 2

**Summary:**

The paper argues that while RL with Verifiable Rewards (RLVR) improve LLM performance on verifiable tasks (math, code, structured QA), headline gains are usually overstated due to three confounders: an “RLVR tax” (side-effects like overconfident hallucinations, instruction-following drift, and expanded safety/privacy attack surface), evaluation pitfalls (budget mismatch, fragile LLM-judge pipelines, calibration drift), and dataset contamination. The authors provide parity-controlled reproductions and a partial-prompt contamination audit showing that several celebrated gaps shrink under clean evaluation. They then propose a “tax-aware RLVR protocol” that co-optimizes accuracy, grounding, and calibrated abstention, and standardizes budgeting, calibration, and provenance checks.

**Strengths:**

1. Clear thesis: The paper cleanly frames three forces that can inflate RLVR claims, taxes, evaluation pitfalls, and contamination. and situates them within the current literature and practice.

2. Substantive diagnosis: The manuscript devotes substantial space to how each inflation source arises in practice (e.g., pass@k without budget parity, fragile LLM-judges, partial-prompt reconstruction), making the critique actionable rather than purely rhetorical.
Balanced stance. The paper acknowledges regimes where RLVR likely broadens capability (not just sharper search) and positions the protocol as compatible with those positive results.

**Weaknesses:**

1. Incrementality / evidence mix. Several observations (budget parity, judge robustness, contamination) echo concerns raised elsewhere; the novelty lies more in synthesis + packaging than in new empirical evidence.

2. The “tax-aware” protocol is promising, but for readers less familiar with RLVR and evaluation, it’s difficult to judge how much each protocol element actually fixes which failure mode and at what cost. There seem to be a lack of table or result supporting the proposed Tax-aware RLVR protocol.

**Questions:**

1. I would like to see a table comparison of ECE (and accuracy) computed (a) on the shared-attempt subset and (b) on each model’s own attempted set. How often do conclusions about “which model is better calibrated” flip between these two views?

2. In Table 3, could similar suffix-reconstruction rates across models be partially explained by short, fact-seeking questions in SimpleQA? For example,  “On what day, month, and year was Algerian artist Mohammed Racim born?” could be capped to “ On what day, month, and year was Algerian artist? Would this result that both strong and weak models fail to infer what the actual question would be, thus bringing similar performance? Would using other dataset be a better choice?

3. Scope of parity issues beyond RLVR. You argue parity/control problems are common in RLVR. Do you observe the same failure modes (budget mismatch, judge brittleness, calibration neglect) in nearby post-training regimes?

---

> ### Author Response · Authors · 2025-11-28
> **Response to reviewer rvSS**
>
> ## Response to Reviewer rvSS
>
> Thank you for the thoughtful review and for engaging with both the thesis and the trade offs.
>
> On the incrementality and evidence mix, we agree that many of the underlying concerns (budget parity, judge robustness, contamination) have appeared before, and that our novelty is primarily:
>
> - putting them under a single RLVR specific lens; and
> - backing that with parity controlled re evaluation and partial prompt audits focused on recent RLVR checkpoints.
>
> In revision we will make that scope explicit so we are not over selling a "brand new" problem, and we will move the most load bearing numerical results (gap table, attempt and calibration behavior, contamination contrasts) into the main text rather than leaving them buried in the appendix.
>
> On the tax aware protocol, we take your point that, as written, it can read like an undifferentiated checklist. In line with our updated position, we will:
>
> - mark a small minimum core (budget parity plus saturation curves, calibration and abstention metrics, at least one contamination probe, basic judge sanity checks) as the main recommendation; and
> - treat the rest as clearly labeled extensions that are motivated by prior work but not empirically vetted as a bundle.
>
> We will also add a compact table that maps each core element to the failure mode it addresses and to the specific measurements in our paper that support including it.
>
> Regarding your specific questions:
>
> - **Shared attempt vs. own attempt ECE.** We agree this is an important stress test of our "attempt inflation" point. In the revision we will add a table comparing ECE and accuracy under both views and report how often the calibration ranking flips. This will let us either sharpen or soften our claims depending on what the numbers show.
> - **SimpleQA as a control in Table 3.** We agree that the short, fact seeking format can compress differences and that our current framing is too optimistic. We will clarify that SimpleQA is used only as a rough non math control (to check that the strong tail reconstruction effect is not ubiquitous), not as a clean or ideal counterexample. We will either, space permitting, add a second, longer non math control or explicitly flag this limitation.
> - **Parity issues beyond RLVR.** Yes, in our own experience the same failure modes (budget mismatch, judge brittleness, calibration neglect) appear in SFT, RLHF, and test time scaling. We will bring this out more clearly and cast RLVR as a convenient testbed for these reliability questions, rather than a setting where they uniquely arise.
>
> We appreciate your suggestions. They helped us sharpen both the claims and the way we present the protocol.

---

### Official Review · Reviewer_zaT4 · 2025-11-02

**Soundness:** 2
**Presentation:** 3
**Contribution:** 2
**Rating:** 2
**Confidence:** 3

**Summary:**

This paper summarized key recent progresses in development of RLVR, specifically focusing on RLVR tax, evaluation pitfalls and data contamination. The author propose a comprehensive tax-aware RLVR protocol to calibrate RLVR methods.

**Strengths:**

1. The author gives a comprehensive review on recent  RLVR studies, pointing the RLVR tax, evaluation pitfall and data contamination problems.
2. The paper is well-written and easy to read.

**Weaknesses:**

1. The technique contribution is not clear, most results are already found in existing literature. The author might have to conduct further deeper analysis.
2. Lacking necessary experiments to demonstrate the effectiveness of the proposed RLVR protocol.

**Questions:**

See weakness.

---

> ### Author Response · Authors · 2025-11-28
> **Response to reviewer zaT4**
>
> Thank you for the clear summary and for pointing out that, as written, the paper leans more toward a survey than a standalone technical contribution.
>
> On the technical contribution and "mostly existing literature" point, we agree that the current draft does not distinguish sharply enough between what we are adding and what we are curating. Our intention is not to claim a new RLVR algorithm, but rather to:
>
> - put three inflation sources (taxes, evaluation pitfalls, contamination) under a single RLVR focused lens;
> - re score a set of recent models under a shared, parity controlled setup (including the partial prompt contamination audit); and
> - distill from those results a small "minimum standard" for RLVR reporting.
>
> In revision, we will make that scope explicit and avoid giving the impression that the contribution is a brand new method.
>
> On the lack of experiments for the protocol, we agree that the current presentation of the protocol over promises. We have not fully validated every knob we mention. Rather than pretending otherwise, we will:
>
> 1. Narrow the claims and present the protocol explicitly as a minimal recommended standard grounded in the specific measurements we do have (budget parity, calibration and abstention behavior, contamination probes); and
> 2. Add a small number of targeted analyses that directly connect those measurements to the most basic controls we advocate.
>
> We will be clear that a full ablation of the entire protocol is out of scope for this position paper, and we will treat the remaining suggestions as hypotheses for future empirical work.

---

### Official Review · Reviewer_ykJK · 2025-11-05

**Soundness:** 1
**Presentation:** 2
**Contribution:** 3
**Rating:** 2
**Confidence:** 3

**Summary:**

This position paper argues that RLVR reliably boosts verifiable tasks (math/code), but headline “reasoning” gains are often overstated due to three confounders:

(i) an RLVR tax (reduced abstention, increased overconfidence, instruction-following drift, larger safety/privacy surface) -- discussed in Section 3
(ii) evaluation pitfalls (budget mismatch, fragile LLM judges, calibration drift) -- discussed in Section 4 and
(iii) data contamination -- discussed in Section 5.

The authors sketch the future as a tax-aware training and evaluation protocol with componentized rewards (correctness/grounding/refusal), calibration-gated early stopping, matched-budget saturation curves, judge-robustness probes, and partial-prompt contamination audits and  parity-controlled reproductions.

**Strengths:**

S1 **Timely, well-scoped problem.** The paper isolates three concrete inflation sources—taxes, evaluation pitfalls, contamination—and keeps a constructive stance: preserve RLVR’s practical wins while fixing reliability and measurement. Fig. 1 clearly maps the agenda.

S2 **Strong coverage of recent work under a common umbrella.** The draft cites key 2024–2025 threads it builds on: pass@k/test-time scaling, judge reliability/surveys, code contamination, budget-parity normalizers (“SoberScore”), GHPO/difficulty-aware schedules, and “think-when-needed” protocols.

S3 **Constructive position.** Many teams can adopt these protocols today.

**Weaknesses:**

W1. **Stance reads non-committal.**
- The paper states too vague a position of “RLVR is valuable and industry-ready, but its benefits are easiest to preserve when reliability, safety, and measurement are treated as first-class objectives.” which could be said for anything. What wouldn't benefit from reliability, safety and measurement?
- Specifically, Section 6 aggregates wide range of controls (budget parity, seed averaging, judge stress tests, contamination hygiene) but don’t isolate which parts matter the most. My primary question: Which of many proposed elements move the needle the most, and by how much? The contribution seem more curatorial than guiding practitioners toward the highest-leverage controls.
- P.S. I’m treating the work as a position paper and ignoring the lack of experiments here that validate the protocols. I leave fit-to-venue to the AC.

W2. **Claim 1 (refusals vs. confident errors) needs isolation.**
- Is this RLVR-specific? SFT on the same data appears to show similar effects (Table 1 and several cited papers). If so, the issue looks like domain or reasoning-training, not an RLVR tax.
- On hallucination/overconfidence: the critique is fair but targets the objective more than RLVR itself. For example, Beyond Binary Rewards: Training LMs to Reason About Their Uncertainty (arXiv:2507.16806) adds an uncertainty loss. But I agree that evaluations should measure this explicitly.

W3 **Claim 2 (privacy/safety surface) lacks a clear threat model.**
- The paper argues that long, explicit traces widen attack and leakage surfaces. Other work finds that traces aid monitorability (e.g., “Chain-of-Thought Monitorability”), which is more plausible to me.
- Maybe the crux of the difference is clarifying who sees traces, when, and how. Are traces exposed via API outputs? If users see only final answers, this vector seems weak. Open-weights models have stronger safety and privacy threat models. When are  RLVR traces the primary concern?

W4 **Claim 3 (evaluation pitfalls) vs. noise.**
- Table 2 shows AIME/AMC gaps often within ±3–5 points apart from 1.5B models excluding nemotron — not too far apart from reported standard deviations (Mu et al., 2025; Hochlehnert et al., 2025). If effects sit in the noise, the original papers may have reported performance correctly. The evidence does not seem to support big mismatches.
- Table 3 suggests models reconstruct hidden tails and final answers in math (a successful reproduction of Wu et al., 2025b) and compares against a SimpleQA control. Qwen models leak more on math, but any leakage is concerning. Is the claim that SimpleQA shows no leakage and the variance stems from metric brittleness? Acc@80 and R@80 look comparable across datasets, which leaves it unclear whether the high contamination score reflects a brittle metric or real contamination in both Math and SimpleQA.

Basically, I worry that the claims do not match the evidence provided, and several issues seem general to reasoning/domain training rather than RLVR. Reported evaluation gaps look small. The contribution reads as a broad checklist without priorities; could the authors identify and justify the few controls that matter most?

**Questions:**

Could the authors address the weaknesses?

---

> ### Author Response · Authors · 2025-11-28
> **Response to Reviewer ykJK**
>
> Thank you for the detailed and concrete feedback. Your review directly motivated the sharpened stance and restructuring summarized in the global response above.
>
> ### W1 – Stance and "curatorial" protocol
>
> We agree that the previous framing ("RLVR is valuable and industry ready...") was too generic. In the revision we will adopt the more concrete thesis given in the global response and place it immediately after the first paragraph of the Introduction.
>
> Section 6 will be shortened and organized around the small minimum standard (budget parity plus saturation curves, calibration and abstention reporting, one contamination audit, and basic judge sanity checks), with the remaining controls clearly labeled as optional extensions rather than implied requirements. We will also state explicitly that we are not trying to rank all controls. Our aim is to argue that this core set is hard to omit if one wants robust RLVR claims.
>
> ### W2 – Is refusal → confident attempts really an "RLVR tax"?
>
> We agree this pattern is not unique to RLVR and can appear with reasoning style SFT on the same data. We will rephrase our claim as a more general "attempt inflation" phenomenon for current reasoning oriented objectives, and make clear that RLVR often amplifies it because of sparse verifiable rewards and entropy pressure.
>
> Table 1's SFT versus RL comparison will be presented explicitly as one concrete instance of this broader effect. We will also reposition uncertainty aware objectives (such as the cited work on training models to reason about their own uncertainty) as natural "tax aware" refinements, rather than treating the issue as intrinsic to RLVR itself.
>
> ### W3 – Safety and privacy surface and missing threat model
>
> We agree that our current discussion of "widened attack surface" is underspecified. We will add a short threat model box in the main text that distinguishes:
>
> - open weights versus API versus internal logs;
> - training versus inference exposure;
> - what "risk increase" concretely means for us (for example higher automated jailbreak success at fixed budgets, easier reconstruction of sensitive content); and
> - when richer traces help via monitorability and safety signals.
>
> In other words, we will frame chain of thought and long reasoning as a trade off: more observable but potentially more exposed when surfaced or reused, rather than as a one sided liability.
>
> ### W4 – Evaluation gaps vs. noise; contamination vs. metric brittleness
>
> We agree that many standardized deltas fall within a typical 3 to 5 point band, especially outside the 1.5B family, and the current wording over suggests "large mismatches". We will soften that language and emphasize instead that parity controls turn some apparent clear wins into smaller or ambiguous differences, rather than claiming dramatic reversals.
>
> For contamination, we will clarify that SimpleQA is used as a domain control rather than a guaranteed "clean" baseline. The main signal we rely on is the contrast, for Qwen style models, between legacy math sets and AIME 25 (high ACC@80 on the former, near zero on the latter), with SimpleQA only checking that strong tail reconstruction does not appear uniformly outside math. We will also add a brief discussion of metric sensitivity so that the limitations of ACC@80 and ROUGE@80 are explicit.
>
> We appreciate the push to make our stance more precise, and we hope the revised framing makes clearer what we are and are not claiming.

---

### Author Response · Authors · 2025-11-28
**Global Response**

## Global response and updated position

We thank all reviewers for thoughtful comments and for pushing us to sharpen both our stance and the scope of our claims. A common thread across the reviews is that
(i) our position reads too generic,
(ii) the "tax aware protocol" feels like a long checklist rather than a focused contribution, and
(iii) the paper does not clearly separate what our own evidence supports from what we extrapolate from the broader literature.

### Updated position

In the revision we will make our thesis explicit, early, and narrower:

> Our position is that RLVR is effective and deployable for verifiable domains, but many headline "reasoning" gains should be treated as provisional unless a small core of controls is enforced: budget parity, calibration and abstention tracking, basic judge stress tests, and a simple contamination audit.

We will also clarify what we mean by an "RLVR tax". We use this term as shorthand for empirical side effects that recur under current reasoning style post training pipelines: reduced abstention, miscalibration, instruction drift, and an expanded safety and privacy surface when long chain of thought and large best of N are pushed aggressively. We do not claim these costs are mathematically inevitable or unique to RLVR. Similar patterns appear for strong SFT on reasoning data. RLVR is our focus because verifiers and open weights make these trade offs measurable.

### From checklist to minimum standard

Section 6 will be restructured around a compact "minimum standard" that is directly grounded in our analyses:

1. Matched budgets with saturation curves.
2. Reporting calibration and abstention alongside accuracy.
3. At least one basic judge robustness probe when LLM judges are used.
4. A partial prompt contamination audit on at least one fresh test set.

Other items we currently mention will be clearly marked as hypothesized extensions rather than part of the core contribution. We will also add a small table that maps each minimum standard item to the failure mode it addresses.

### Evidence vs. speculation

Several reviewers asked which controls "move the needle" and by how much. Our current experiments already show that:

- Budget parity and template control compress or flip some gaps, especially for 1.5B models.
- Attempt inflation and calibration shifts can significantly change how one ranks models.
- Contamination patterns differ sharply between legacy math and fresh math benchmarks.

Within rebuttal constraints, we will add two small, focused analyses:

1. A comparison of accuracy and ECE computed
   (a) on the shared attempt subset versus
   (b) on each model's natural attempted set,
   to make the impact of attempt inflation and calibration drift concrete; and

2. An extended contamination analysis that contrasts legacy math benchmarks, AIME 2025, and at least one non math control, including a short discussion of metric sensitivity (ACC@80 and ROUGE@80 versus alternative views).

These are not full ablations of every protocol knob, but they tighten the link between our empirical sections and the minimum standard we recommend. More broadly, we will be explicit that we have not optimized or ranked all possible controls. The intended contribution is to argue that this small core is hard to omit if one wants credible RLVR claims.

We respond to each reviewer's specific comments below.

---

### Note · Authors · 2026-01-03

**Comment:**

We would like to thank all reviewers for the helpful suggestions.

**Withdrawal Confirmation:**

I have read and agree with the venue's withdrawal policy on behalf of myself and my co-authors.